



# Is the ocean surface a source of nitrous acid (HONO) in the marine boundary layer?

**Leigh R. Crilley[1,a], Louisa J. Kramer[1,b], Francis D. Pope[1], Chris Reed[2,c], James D. Lee[2], Lucy J. Carpenter[2], Lloyd D. J. Hollis[3], Stephen M. Ball[3], and William J. Bloss[1]**

[1.] *School of Geography, Earth and Environmental Sciences, University of Birmingham, Birmingham, United Kingdom*

[2.] *Wolfson Atmospheric Chemistry Laboratory (WACL), Department of Chemistry, University of York, Heslington, York, United Kingdom*

[3.] *School of Chemistry, University of Leicester, Leicester, United Kingdom*

[a.] *now at: Department of Chemistry, York University, Toronto, ON, Canada*

[b.] *now at: Ricardo Energy & Environment, Harwell, Oxfordshire, UK.*

[c.] *now at: FAAM Airborne Laboratory, Building 146, Cranfield University, Cranfield, UK*

*Correspondence to*: Leigh R Crilley (lcrilley@yorku.ca)

**Abstract.** Nitrous acid, HONO, is a key net photolytic precursor to OH radicals in the atmospheric boundary later. As OH is the dominant atmospheric oxidant, driving the removal of many primary pollutants and the formation of secondary species, a quantitative understanding of HONO sources is important to predict atmospheric oxidising capacity. While a number of HONO formation mechanisms have been identified, recent work has ascribed significant importance to the dark, ocean-surface mediated conversion of $NO_2$ to HONO in the coastal marine boundary layer. In order to evaluate the role of this mechanism, here we analyse measurements of HONO and related species obtained at two contrasting coastal locations – Cape Verde (Atlantic Ocean), representative of the clean remote tropical marine boundary layer, and Weybourne (United Kingdom), representative of semi-polluted Northern European coastal waters. As expected, higher average concentrations of HONO (70 ppt) were observed in marine air for the more anthropogenically influenced Weybourne location compared to Cape Verde (HONO <5 ppt). At both sites, the approximately constant HONO/$NO_2$ ratio at night pointed to a low importance for the dark ocean-surface mediated conversion of $NO_2$ into HONO, whereas the midday maximum in the HONO/$NO_2$ ratios indicated significant contributions from photo-enhanced HONO formation mechanisms (or other sources). We obtained an upper limit to the rate coefficient of dark ocean-surface HONO-to-$NO_2$ conversion of $C_{HONO} = 0.0011$ ppb hr$^{-1}$ from the Cape Verde observations; this is a factor of 5 lower than the slowest rate reported previously. These results point to significant geographical variation in the predominant HONO formation mechanisms in marine environments and indicate that caution is required when extrapolating the importance of such mechanisms from individual



study locations to assess regional and/or global impacts on oxidising capacity. As a significant fraction of atmospheric
processing occurs in the marine boundary layer, particularly in the tropics, better constraint of the possible ocean surface
source of HONO is important for a quantitative understanding of chemical processing of primary trace gases in the global
atmospheric boundary layer and associated impacts upon air pollution and climate.

## 1 Introduction

Chemical processing moderates the link between emissions and abundance for many atmospheric constituents, affecting
climate, biogeochemical cycling and health impacts. The dominant atmospheric oxidant is the hydroxyl (OH) radical. OH
initiates the daytime degradation of many constituents including air pollutants such as $NO_2$, volatile organic compounds
(VOCs), and greenhouse gases such as methane. This chemistry forms secondary species including ozone and organic and
inorganic aerosol particles, which are harmful to human health and the environment, and affect radiation transfer and hence
climate. Consequently, quantitative understanding of the OH budget and hence the abundance of OH precursor species is of
critical importance for atmospheric science.

In the atmospheric boundary layer, nitrous acid (HONO) is an important photolytic precursor to OH production (R1).
HONO can be formed in the gas-phase via the reaction between NO and OH (R2); in the absence of other HONO formation
or loss mechanisms or direct HONO emission sources, R1 and R2 can reach a photo-stationary steady state (PSS), with
HONO representing an OH (and NO) reservoir. Reaction with OH (R3) represents a further sink for HONO but is considered
to be minor compared to photolysis (R1) under typical tropospheric (daytime) conditions.

$HONO + hv$ (λ < 400nm) → NO + OH                                                                              (R1)

NO + OH + M → HONO + M                                                                                        (R2)

$OH + HONO → H_2O + NO_2$                                                                                      (R3)

During the day at mid-latitudes, HONO has a relatively short atmospheric lifetime with respect to photolysis (in the order of
tens of minutes) and yet numerous field studies report non-negligible daytime HONO concentrations (see e.g. (Kleffmann et
al., 2005; Lee et al., 2016; Michoud et al., 2014; Pusede et al., 2015; Villena et al., 2011)). Calculations of daytime HONO
concentrations based on the photo-stationary steady state (R1-3) significantly underestimate the observed daytime HONO
concentrations pointing to the existence of other unknown HONO sources or formation mechanisms (Crilley et al., 2016;
Michoud et al., 2014; Sörgel et al., 2011; VandenBoer et al., 2013). This overall conclusion is robust to the challenges of
applying PSS in study locations with significant spatial heterogeneity of sources and consequent lack of photochemical
equilibrium (Crilley et al., 2016; Lee et al., 2013), and additional HONO formation mechanisms (R4) are commonly required
to satisfactorily reproduce observed HONO levels. These sources lead to further HONO production (above and beyond R2),
and hence net OH (and NO) formation, following R1.

Source → HONO                                                                                              (R4)



A number of additional sources of HONO (R4) have been identified in the literature including: direct emissions (e.g. vehicles); biological activity (e.g. soil bacterial activity); possible homogenous gas-phase reactions; heterogeneous reactions on aerosol, the ground and vegetation; and surface photolysis (see reviews by (Kleffmann, 2007; Spataro and Ianniello, 2014)). However, an overall understanding of the source strengths and mechanisms remains elusive, and multiple HONO

production mechanisms are likely to occur in parallel, to differing extents, in contrasting environments.

The contribution of HONO to the primary OH budget may be readily evaluated from atmospheric field observations, and in continental boundary layer locations has been shown to range from 33% in forested areas (Kleffmann et al., 2005) to 80.4% in semi-rural locations (Kim et al., 2014).  The contribution of HONO has also been shown to be significant in urban areas, ranging from 40-83% of the primary OH budget (Emmerson et al., 2005; Lee et al., 2016; Ren et al., 2006; Slater et al.,

2020). These figures relate to primary OH production, i.e. neglecting radical recycling via e.g. $HO_2 + NO$.

Oceans account for more than 70% of the Earth's surface. As a result, the marine boundary layer (MBL) is an important environment globally when considering oxidation; for example the marine lower troposphere has been shown to account for 25% of the global chemical sink of methane (Bloss et al., 2005). There are however only very limited measurements of HONO within the marine boundary layer reported in the literature to date. Vecera et al. (2008) performed measurements on a

ship cruise in the Aegean Sea and observed HONO concentrations that were typically low (<50 ppt), except in fresh ship plumes (<10 ppb). Ye et al. (2016) conducted aircraft measurements over the North Atlantic ocean and reported average HONO levels within the marine boundary layer of 11.3 ± 1 .6 ppt. Low levels of HONO were also reported by Reed et al. (2017) at a coastal site in Cape Verde, representative of the remote marine boundary layer, where HONO peaked at an average of 3.5 ppt. In contrast, other studies report much greater HONO concentrations: for example Cui et al. (2019)

reported shipboard HONO measurements from an area with heavy shipping traffic (East China Sea), at an average of 480 ± 210 ppt within 30 km of the coastline, decreasing slightly to 400 ± 180 ppt at distances up to 100 km.  Land-based measurements from Chinese coastal sites report a similar range of abundance:  Zha et al. (2014) reported mean mixing ratios of 126 ± 95 ppt and 4.06 ± 3.29 ppb for HONO and $NO_2$ respectively at a clean coastal site near Hong Kong. Similar levels were reported for Tuoji island in the eastern Bohai Sea, China by Wen et al. (2019) of 200 ± 200 ppt and 5.3 ± 4.1 ppb for

HONO and $NO_2$ respectively.  Wojtal et al. (2011) observed high concentrations of HONO (500-1500 ppt) and $NO_2$ (<50 ppb) during night-time measurements at a coastal site that experienced regular outflow of air pollution from nearby Vancouver, Canada.

The scarcity of HONO measurements within the marine boundary layer makes it difficult to ascertain "typical" levels, but the studies to date point to HONO abundance within the ppt range at mid-latitudes (Cui et al., 2019; Večeřa et al., 2008; Wen

et al., 2019; Ye et al., 2016; Zha et al., 2014; Reed et al. 2017). Owing to this paucity of HONO measurements, estimates of the contribution of HONO to the OH budget in the MBL are limited. Calculations by Cui et al. (2019) showed average daytime OH production from HONO photolysis (1.35 ± 0.69 ppb hr$^{-1}$) was 1.6 times that calculated from $O_3$ photolysis (daytime average levels of 300-600 ppt for HONO and 40-70 ppb and for $O_3$), suggesting an important role for HONO photolysis in the oxidising capacity of the MBL.





Recent work has presented indirect evidence for rapid production of HONO on the ocean surface. Based on night-time field measurements at coastal sites around the Bohai and South China seas, Zha and co-workers (2014) observed that the $HONO/NO_2$ ratio increased more rapidly in air masses which passed over the sea, compared with those which passed over land. Assuming that all the measured HONO derived solely from $NO_2$ conversion, this observation implied a higher $NO_2$ to HONO conversion rate over the sea compared to the land, and this was attributed to the existence of a source of HONO from

the ocean's surface. Enhanced nocturnal $NO_2$ conversion to HONO for marine air relative to land was also observed in subsequent studies in these locations (Cui et al., 2019; Wen et al., 2019), and was again attributed to heterogeneous processes occurring on the ocean surface.

This result is somewhat counter-intuitive, as HONO ought to be highly soluble in seawater (due to seawater being slightly alkaline, pH of 8) and would dissociate in solution to $H^+$ and $NO_2^-$ ions. Therefore, one would expect the ocean surface to be

a net sink of HONO. However, the sea surface organic microlayer may inhibit contact of gas-phase species with the bulk seawater, restricting the ability of the ocean surface to act as a HONO sink – rather, the surface microlayer may indeed provide a medium or surface for HONO formation reactions.

Heterogeneous processes are a prominent HONO formation mechanism (Kleffmann, 2007), occurring mainly via the dark conversion of $NO_2$ on wet surfaces (Finlayson-Pitts et al., 2003):

$2NO_2 + H_2O \rightarrow HONO + HNO_3$                                  (R5)

HONO can also be formed by photo-enhanced reduction of $NO_2$ on organic substrates (George et al., 2015; Stemmler et al., 2006):

$NO_2 + HC_{red} \rightarrow HONO + HC_{ox}$                                  (R6)

Organic films comprising humic acids have been shown to support HONO production (Bartels-Rausch et al., 2010;

Stemmler et al., 2006) and the sea surface microlayer of organic material is known to contain humic acids (Williams et al., 1986). As a result, the photo-reduction of $NO_2$ on the sea surface organic microlayer (via R6) may be a potential source of HONO in the marine boundary layer (Wen et al., 2019; Wojtal et al., 2011; Yu et al., 2021). It seems likely that HONO production is driven by the ocean's surface, rather than aerosol particles, because the ocean surface area is typically much larger than aerosol surface area (e.g. 15-fold during the day, based on a typical particle surface area for the marine boundary

layer of 150 $\mu m^2 \, cm^{-3}$ (VandenBoer et al., 2013) and a daytime boundary layer height of 1000 m; this ratio is even larger at night). Aerosols can be a source of HONO in the marine boundary layer via the rapid photolysis of nitrate-containing particles; however there is significant uncertainty to the importance of this route, with reported rates spanning three orders of magnitude (Kasibhatla et al., 2018; Reed et al., 2017; Shi et al., 2021; Ye et al., 2016). Zhang et al. (2016) included a parameterisation for an enhanced $NO_2$ to HONO conversion rate over the sea surface based upon the results of Zha et al.,

(2014) and calculated that this marine source accounted for 9% of the observed HONO mixing ratio in Hong Kong (implying a significant influence over OH production). The results from these studies (Zha et al., 2014; Zhang et al., 2016) suggest the ocean surface may be a significant source of HONO in some coastal and marine areas.





In the current work, we evaluate the potential role of the ocean surface as a source of HONO into the marine boundary layer through the analysis of ground level field measurements of HONO performed at two contrasting coastal locations. The two sites are: the Cape Verde Atmospheric Observatory (CVAO) in the tropical North Atlantic ocean, representative of the global tropical remote marine boundary layer (Carpenter et al., 2010), and the Weybourne Atmospheric Observatory (WAO) on the east coast of the United Kingdom, representative of semi-polluted marine boundary layer environments around Europe (Forster et al., 2012).

## 2 Methods

### 2.1 Measurement of HONO

Nitrous acid (HONO) was measured at each site using a Long Path Absorption Photometer (LOPAP-03 QUMA Elektronik & Analytik GmbH), as described in detail in Heland et al., (2001) and Kleffmann et al., (2002). Briefly, the LOPAP is a wet chemical technique where gas-phase HONO is sampled within a stripping coil into an acidic solution where it is derivatized into an azo dye. The absorption of light at 550 nm by the azo dye is then measured with a spectrometer using an optical path length of 2.4 m. The LOPAP was operated and calibrated according to the standard procedures described in Kleffmann and Wiesen (2008) at both sites. Further details on the operation of the LOPAP at CVAO can be found in Reed et al. (2017). The resultant time resolution of the LOPAP was 5 minutes and baseline measurements were taken at frequent intervals (every 6-8 hrs) at each site. The detection limit ($2\sigma$) was determined to be 0.2 ppt at CVAO and 0.7 ppt at WAO. The same LOPAP instrument was used at each site and was operated according to the same procedures, this should remove any variability that might otherwise arise when comparing measurements made by different instruments (Crilley et al., 2019; Pinto et al., 2014). The two-channel approach of the LOPAP has also been shown to be necessary in clean environments where the HONO concentration is expected to be low, as it successfully accounts for possible interferences (Kleffmann and Wiesen, 2008).

### 2.2 Site Descriptions

#### 2.2.1 Remote Marine: Cape Verde Atmospheric Observatory

The Cape Verde Atmospheric Observatory (CVAO) is located in the tropical North Atlantic (16.864°N, −24.868°E) on the northwest coast of the island of São Vincente and is a global World Meteorological Organisation (WMO) Global Atmospheric Watch (GAW) station. CVAO is considered representative of the clean marine boundary layer with typical sampled air masses originating from either the open Atlantic Ocean or Saharan Africa (Carpenter et al., 2010). At CVAO, long-term routine measurements of NOx, O₃, meteorology and photolysis rates are performed, with further details of methodology and climatology presented in Lee et al. (2009) and Andersen et al. (2020). In addition to these routine measurements, for this study in November 2015, the concentration of HONO was measured with a sampling height of approx. 3 m above ground level.





### 2.2.2 Semi-polluted marine: Weybourne Atmospheric Observatory

The Weybourne Atmospheric Observatory (WAO) is a regional Global Atmospheric Watch station located on the north
Norfolk coast, United Kingdom (52.95°N, 1.13°E) and has been described in detail by Penkett et al. (1999). WAO is
approximately 170 km northeast of London and experiences air masses from a number of sources including relatively clean
maritime air and pollution outflow from the UK and continental Europe (Forster et al., 2012). Measurements were
conducted in July 2015 as part of the Integrated Chemistry of Ozone in the Atmosphere (ICOZA) project, which aimed to
explore the chemistry of ozone production. During ICOZA, the LOPAP sampled at a height of approx. 3m above ground
level. Measurements of NO were performed by chemiluminescence using a calibrated NOx (Air Quality Designs inc)
monitor described in Reed et al. (2016). Absolute spectroscopic measurements of $NO_2$ were performed by broadband cavity-
enhanced absorption spectroscopy in the wavelength range 430-485nm (typical $1\sigma$ detection limit = 0.035 ppbv in 1 minute
in the presence of ambient aerosol) (Thalman et al., 2015). The $NO_2$ inlet line sampled air at approx. 4.7 m above ground
level. A spectral radiometer (MetCon GmbH, Germany) was used to measure the photolysis rates of some 40 photolabile
molecules including $NO_2$, HONO, and $O_3 \rightarrow O(^1D)$ (Edwards and Monks, 2003). In addition, routine measurements of the
meteorology, gaseous species and particle measurements were performed at WAO. In this study we utilise measurements of
$SO_2$ (fluorescence, Thermo 43i), ozone (UV photometry, Thermo 49i) and particle concentration (TSI 3025a).

### 2.3 Data Analysis

Wind rose plots, diurnal cycles and polar plot analyses were performed using the Openair package (Carslaw and Ropkins,
2012) in the R software package.

### 3.0 Results and Discussion

### 3.1 Comparison of measured concentrations: WAO vs CVAO

The measured NOx and HONO mixing ratios from each site are summarised in Table 1 and presented as measurement time
series in Figures 1 (CVAO) and 2 (WAO). In both cases significant, but contrasting, variability was observed, with levels of
both NOx and HONO much higher for WAO than CVAO, by factors of 60 and 80 respectively (mean mixing ratios
observed during periods of marine air). These differences reflect the remote, unpolluted nature of Cape Verde, and proximity
of the Weybourne site to a wide range of anthropogenic emissions from the UK and continental Europe. Overall, despite the
higher absolute concentrations of NOx and HONO at WAO compared to CVAO, the HONO/$NO_2$ ratios were remarkably
consistent between the two sites, suggesting some degree of commonality in behaviour in the chemical NOy space.


**Table 1: Mean mixing ratios of HONO, NO, NO₂ and O₃ at WAO (July 2015) and CVAO (25 Nov- 3 Dec 2015). Ozone diurnal profiles are provided in SI (Figs S1 and S2). WAO data including averages over the whole measurement period ("all") and subdivided into periods when the local wind direction was from the ocean (NW, N, NE: "marine", *ca*. 23% of all data) or other sectors ("non-marine", predominantly SE-SW). Averages correspond to periods for which measurements of all four species were available simultaneously; uncertainty is 1 sd of the observed levels. The HONO/NO₂ ratios were determined by linear regression between all data points (Fig S3).**

| | HONO (ppt) | NO (ppt) | NO₂ (ppt) | O₃ (ppb) | HONO/NO₂ |
|---|---|---|---|---|---|
| CVAO | $0.84 \pm 1.0$ | $6.4 \pm 25$ | $38 \pm 25$ | $39 \pm 2$ | $0.025 \pm 0.03$ |
| WAO all | $90 \pm 87$ | $398 \pm 607$ | $2103 \pm 1527$ | $33 \pm 15$ | $0.024 \pm 0.001$ |
| WAO marine | $70 \pm 78$ | $472 \pm 700$ | $2131 \pm 2019$ | $33 \pm 7$ | $0.023 \pm 0.002$ |
| WAO non-marine | $98 \pm 91$ | $387 \pm 587$ | $2142 \pm 1362$ | $33 \pm 14$ | $0.022 \pm 0.001$ |


The local wind direction at Cape Verde during the campaign period was wholly from the NE quadrant, and overwhelmingly from the NE / ENE directions (See SI Figure S4), corresponding to the open ocean sector from the shoreline observatory's location. Back trajectory analysis confirmed that the air mass was from the open ocean (Fig S5, Supporting Information). A consistent diurnal variation in HONO was observed, with the exception of the 28th November 2015. The NO and NO₂

concentrations (but not HONO measurements) were characterised by a number of sharp spikes, thought to be shipping plumes (Lee et al., 2009) and with the exception of these plumes, the observed NOₓ levels showed little diurnal variability (Fig 1 and left panels of Fig 3). Measured NO and NO₂ concentrations in the current work were in broad agreement with previous measurements at CVAO for the corresponding time of year (Lee et al., 2009; Read et al., 2008).

At Weybourne, the concentrations of both HONO and NOx demonstrated a large variability, with the time series spanning

periods of generally high levels early in the time series, and lower mean and peak concentrations later in the dataset. Air masses were classified as marine for the NW – NE local wind sector, corresponding to the direction of the North Sea relative to the sampling site, and accounted for 23% of the measurements; during these periods, the mean concentrations of HONO were lower, relative to the non-marine period (Table 1 and Fig 2). For a significant portion of the campaign (namely 29th





June to the 10th July), back trajectory analysis indicated that air masses were mainly from south-west (see example

trajectories in Fig S6) and therefore in the pollution outflow from London (Forster et al., 2012) which likely explains the

higher HONO and NOx concentrations. During periods of marine air, back trajectory analysis indicated that the air masses

travelled over the ocean for 6 hrs prior to arriving at the sampling site, but before this had passed over land in the Northern

UK (Fig S7). For the remainder of the analysis presented here, for the Weybourne dataset, we focus only on times when the

local wind direction was from the ocean (NW, N and NE), referred to as WAO marine.

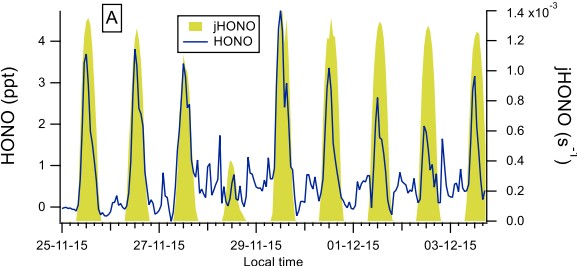


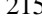

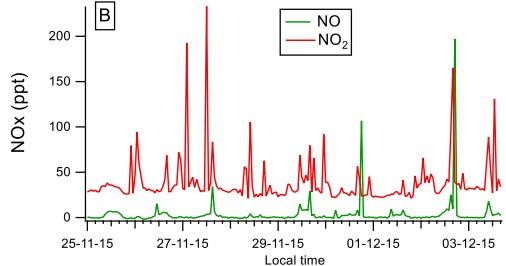

**Figure 1: Time series of the measured HONO, jHONO (A) and NOx (B) mixing ratios at CVAO.**



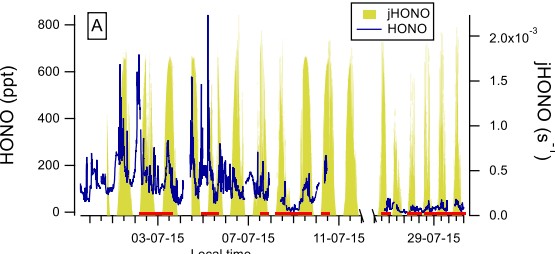


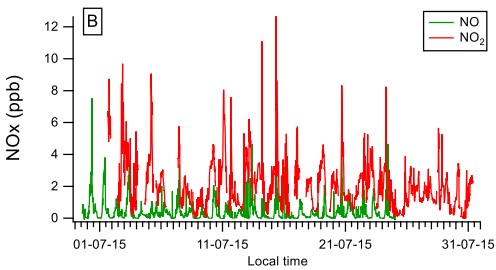

**Figure 2: Time series of the measured HONO and jHONO (A) and NOx (B) mixing ratios at WAO. Red bars on the HONO time series x axis indicate marine air periods at WAO. Note the larger vertical axis scales indicating greater concentrations of HONO and NOx were observed at WAO than in the CVAO data shown in Figure 1.**

## 225 3.2 Diurnal trends in HONO and NOx

Distinct diurnal profiles were observed for HONO and NO (Fig 3). At CVAO, both HONO and NO were very low at night, and peaked during the day at around local solar noon at 12:00 (Fig 3). At midday the lifetime of HONO with respect to photolysis was short (~13 minutes) and, in the presumed absence of any local emission sources and because these data are dominated by marine air masses, the observed diurnal cycle suggests a significant photolytic source term for HONO (see

section 3.3 below). In contrast, the $NO_2$ levels at CVAO showed almost no diurnal variation, with any deviations from the mean being due to spikes of $NO_2$ from advection of (distant) shipping plumes (Andersen et al., 2020).





Figure 3 also presents diurnal trends for marine air at WAO, which were broadly consistent with the observed diurnal trends for the non-marine period (Fig S8, Supporting Information). At WAO a complex diurnal profile was observed, with peak HONO concentrations occurring at night (midnight to 5am) and mid-morning (9:00) and a minimum at 18:00 during late

afternoon (Fig 3), whereas $NO_2$ levels peaked overnight.  The increasing concentration of HONO during the darkness from 20:00 (sunset) to 06:00 (sunrise) points to a sustained, non-photolytic source term.

The absolute ratios of $HONO/NO_2$ were similar at both sites (Table 1) along with similar diurnal trends observed (Fig 4). A pronounced peak in $HONO/NO_2$ was observed at CVAO in the middle of the day. The highest HONO/NO2 was also seen around the middle of the day at WAO, although the diurnal profile was more structured (the drop observed at 12:00 for

WAO marine air is likely due to unfortunately limited data coverage at this hour). At both sites, the $HONO/NO_2$ ratio was relatively constant at night although the absolute values differed: the mean ratio calculated between 20:00-06:00 was 0.012±0.004 and 0.033±0.005 for CVAO and WAO marine, respectively.

Neglecting any effects due to changing boundary layer height, if there were appreciable dark conversion of $NO_2$ to HONO on the sea surface, then we would expect to observe a steady increase in $HONO/NO_2$ during the night.  This is not observed:

the $HONO/NO_2$ ratio for marine air sectors is approximately constant (slopes (±1 sd) of -0.0003±0.0003 and 0.0007±0.0008 for CVAO and WAO marine, respectively) at night for both sites (Fig 4).  In the following sections, we place these qualitative discussions of marine HONO formation and ocean surface production on a more quantitative basis.



## CVAO          WAO

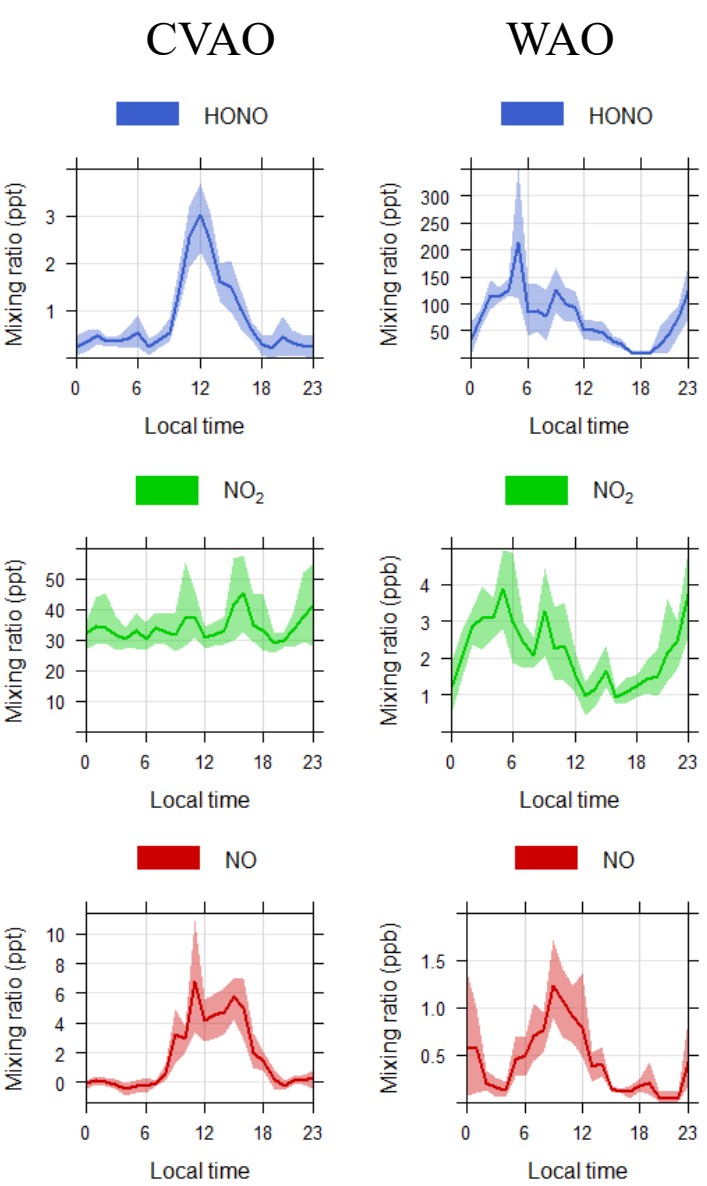


**Figure 3: Diurnal profiles of HONO, NO₂ and NO at CVAO (left column) and WAO marine (right). Shaded areas represent the 95% confidence intervals of the measurement precision. Note the very different y-axis scales.**





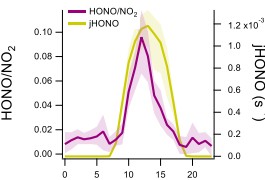

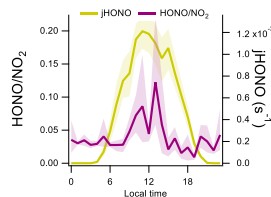

**Figure 4: Diurnal profiles of HONO/NO₂ ratios and HONO photolysis rates at CVAO (left) and WAO marine (right). Shaded areas represent the 95% confidence intervals of the measurement precision.**

### 3.3 Are ocean surfaces a source of HONO?

### 3.3.1 Clean marine environment (CVAO)

At CVAO, the wind was consistently from the ocean (Fig S4), and therefore dark $NO_2$ to HONO conversion on ocean
surfaces, if it occurs at the rates suggested in recent literature studies (Wen et al., 2019; Wojtal et al., 2011; Zha et al., 2014), would be expected to result in an accumulation of HONO overnight (in the absence of ventilation effects). Zha et al. (2014) report an ocean surface dark $NO_2$-to-HONO conversion rate ($C_{HONO}$) of 0.033 hr$^{-1}$, and parameterised MBL HONO production via Eqn 1:

$$\frac{d[HONO]}{dt} = C_{HONO}[NO_2] = 0.033[NO_2] \qquad (1)$$

Using the mean nightly $NO_2$ mixing ratio at CVAO (35 ppt, Fig 3), Eqn 1 produces a HONO growth rate of 1.2 ppt hr$^{-1}$, and minimum HONO levels at dawn (after 12 hrs of darkness) of around 14 ppt. This is not consistent with the observations if HONO were increasing at the 1.2 ppt/hr rate implied by the $C_{HONO}$ value of Zha et al. (2014), then we would expect the HONO concentration to exceed the approx. constant, observed [HONO] = 0.3 ± 0.1 ppt by a factor of 4 within the first hour after sunset and to continue increasing through the night. Wen at al. (2019) reported that when the ambient night-time
temperature was above 15°C, the $C_{HONO}$ was equal to or higher than 0.02 hr$^{-1}$ (but lower than assumed in Eqn 1). During the measurement period at CVAO, the mean ambient night-time temperature was 24±0.5°C. Substituting $C_{HONO}$ = 0.02 hr$^{-1}$ (as a lower limit) into Eqn 1 implies a growth in HONO mixing ratio of 0.7 ppt hr$^{-1}$ (for an $NO_2$ level of 35 ppt), also still significantly in excess of the observations. The observed HONO and $NO_2$ levels display no statistically significant trend with time overnight (d[HONO]/dt = -0.0003±0.0003 ppt hr$^{-1}$) within precision, the CVAO HONO data are consistent with an
upper limit of $C_{HONO}$ < 0.0011 hr$^{-1}$, obtained by applying the methodology of Zha et al. (2014) to the mean HONO measured between midnight and 6 am shown in the diurnal plot in Fig 3. In comparison, Wen et al. (2019) calculated $C_{HONO}$ for 8 nights with stable meteorology in the MBL over the Bohai Sea, China and determined values ranging between 0.006 and 0.036 hr$^{-1}$ (average of 0.018 hr$^{-1}$). These values are still between ×6 and ×33 greater than the upper limit on $C_{HONO}$ we obtain





from the CVAO dataset and are not consistent with the observed (near constant) temporal behaviour in HONO overnight.

This suggests there was little $NO_2$ to HONO conversion occurring on the ocean surface around Cape Verde at night.

To explore the impact of $NO_2$ conversion to HONO at the rates proposed by Zha et al. (2014) during daylight, we extended Eqn 1 to incorporate HONO photolysis and hence estimate the daytime steady-state HONO concentration (excluding any other processes). The resulting equation 2 uses measured $NO_2$ levels and photolysis HONO frequencies derived from the spectral radiometer measurements at Cape Verde:

$$\frac{d[HONO]}{dt} = C_{HONO}[NO_2] - j_{HONO}[HONO] = 0 \quad \rightarrow \quad [HONO]_{ocean} = \frac{C_{HONO}[NO_2]}{j_{HONO}} \qquad (2)$$

Using a value of 0.033 hr$^{-1}$ for $C_{HONO}$, we obtain a daytime minimum for $[HONO]_{ocean}$ of 0.23 ppt (Fig S9, Supporting Information), well below the levels observed at CVAO during the day and with the opposite diurnal behaviour. We conclude that for NOx levels found in the remote / clean tropical marine environment, as represented by the Cape Verde data, the proposed ocean surface conversion mechanism is unlikely to be an important source of HONO at the rate found in the East

China Sea.

### 3.3.2 Semi-polluted marine environment (WAO)

An equivalent analysis was performed for the semi-polluted marine environment using the Weybourne (WAO) dataset. The air mass history is more complex at this location (more variable meteorology), as illustrated in CPF polar plots for HONO and $NO_2$ (Figure 5). Owing to the short atmospheric lifetime of HONO during daylight, we focus on local meteorology (i.e.

wind direction), as this is more relevant when considering local sources of HONO rather than e.g. back trajectories. We did not use this approach for CVAO because the wind and trajectories were consistently from the NE / open ocean sector (Fig S4).

Polar plots indicate that the highest source regions for both HONO and $NO_2$ correspond to southerly / south-westerly directions and higher wind speeds, pointing to regional influence from inland conurbations (ultimately, London). The short

lifetime of HONO in daylight (10 mins at SZA = 40°) means the high concentration of HONO observed in the S and SW wind directions was more probably the result of an increased availability of precursors in pollution outflow (e.g. $NO_2$ reactant for R5 and R6), rather than long-range transport of HONO itself. Similarly, regional emissions of HONO from e.g. road traffic (Kramer et al., 2020) and/or near-site terrestrial sources (see Section 4.0) are likely unimportant. The polar plots for HONO remain broadly similar (Fig S10, Supporting Information) when the data are divided into daytime and night-time

data, again indicating that the land (and not the ocean) was the dominant HONO source at WAO.





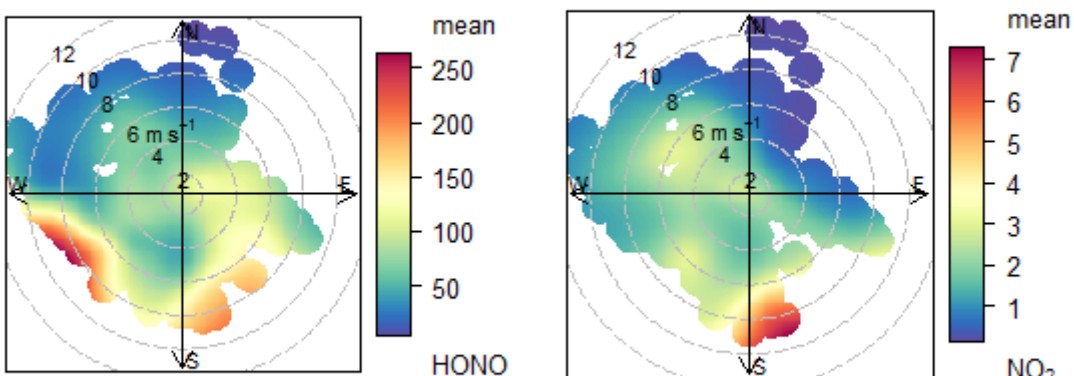

**Figure 5: Polar plot of mean HONO (left, ppt) and NO₂ (right, ppb) mixing ratio for whole time series at WAO (marine + non-marine). The coast is towards the north of the measurement site.**

Non-negligible concentrations of HONO were observed at WAO when the wind was from the ocean sector. Following the

methodology of Zha et al. (2014) and Wen et al. (2019), we examined the measurement time series to look for nights where

the wind was consistently from the ocean and the HONO concentration was observed to increase overnight. One such night

in the WAO dataset met these conditions (2nd/3rd July).  While the HONO concentration was observed to increase overnight,

the HONO/NO₂ ratio was relatively stable (0.02-0.025; Fig 6) because the NO₂ mixing ratio was also increasing; this stable

HONO/NO₂ ratio does not suggest any net NO₂-HONO conversion, in the absence of ventilation effects.





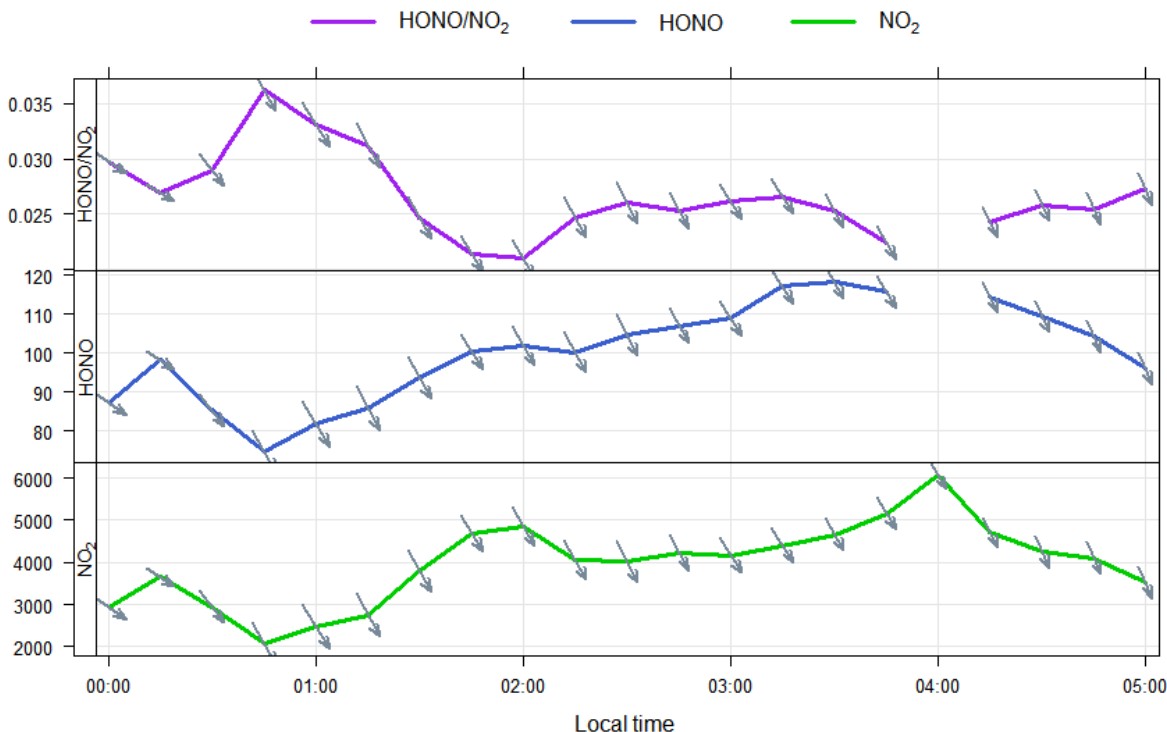


**Figure 6: Time series of HONO/NO₂, HONO (ppt) and NO₂ (ppt) at WAO on the night of 2ⁿᵈ-3ʳᵈ July. The arrows indicate the local wind direction – the marine sector spans WNW – ENE from the WAO station. The break in the HONO time series at 04:00 is due to LOPAP auto- zeroing.**

The increasing HONO concentration observed in the case study of the night of 2ⁿᵈ-3ʳᵈ July may point to some ocean HONO

production. Thus, we applied Eqn 1 to calculate the night-time HONO levels that would result from the proposed ocean

surface conversion mechanism using the [NO₂] measured on this night and the full range of values for $C_{HONO}$: 0.033 (Zha et

al., 2014), 0.018 (Cui et al., 2019) and 0.0011 hr$^{-1}$ (CVAO, current work - upper limit). Fig 7 compares the calculation's

results with the observations. The $C_{HONO}$ values of Zha et al. and Cui et al. substantially over-predict the observations; the

significantly smaller upper limit value derived from the Cape Verde observations is consistent with the measured data but

indicates a minimal contribution to HONO formation from the ocean-surface mechanism at WAO.  While Cape Verde is a

remote location with very low NOₓ, the NO₂ levels at WAO during the case study night (mean of 4 ppb) were similar to

those observed by Zha et al.  (2014) and Cui et al. (2019), at approx. 1 to 3 and 4 to 7 ppb, respectively.





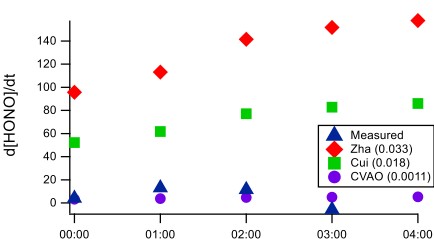

**Figure 7: Measured (blue) night-time HONO production, d[HONO]/dt (ppt hr⁻¹), during the 3ʳᵈ July case study night at WAO, compared to calculations via Eqn 1 for a range of $C_{HONO}$ values.**

Next, we applied the steady-state Eqn 2 to estimate daytime HONO formation at WAO through the ocean-surface mechanism alone. For periods of marine air only and assuming $C_{HONO} = 0.033$ hr⁻¹, we obtain HONO mixing ratios of 8 to 27 ppt between 7am and 4pm, reaching a photolysis-driven minimum at 1pm (Fig S9, Supporting Information). This does not match the shape of the diurnal profile of the HONO measurements (Fig 3), which show a midday peak; although the upper end of the calculated HONO levels from the possible ocean source are the same order of magnitude as the observed HONO (17-80 ppt, Fig 3). This would suggest that, within the constraints of the WAO data, there is scope for some daytime $NO_2$-to-HONO conversion over the sea surface, if a different, possibly photochemical, mechanism also operates, as suggested by Cui et al. (2019).

## 4 Spatial variability and the role of other HONO Sources

The results reported here were obtained from locations (Cape Verde, Weybourne) rather distinct to the Bohai / South China Sea environments studied in previous work (Cui et al., 2019; Wen et al., 2019; Zha et al., 2014), although the levels of key precursor species (NOx) are similar for WAO. The results reported here are in agreement with recent laboratory work by Yu et al (2021), who observed little uptake of $NO_2$ on bulk sea water under dark conditions ($\gamma = 1.6 \times 10^{-8}$); however this may not take into account any effect of the organic sea surface microlayer (SML) on $NO_2$ heterogeneous conversion. Variability in the observed HONO/$NO_2$ behaviour overnight may reflect geographical differences in the organic composition of the SML, which is influenced by underlying surface water composition, biological activity and anthropogenic inputs (Stolle et al., 2020; Wurl and Obbard, 2004). Further work is required to verify whether the SML does or does not facilitate conversion of NO2 to HONO and hence parameterise the variability in the potential night-time $NO_2$-HONO conversion rate on the ocean surface.





A possible alternative marine source of HONO could be shipping emissions. Marine traffic is a significant emitter of NOx (6.5 Tg N year$^{-1}$, 78% of traffic emissions (Eyring et al., 2005)) and HONO has been observed in shipping plumes (Cui et al., 2019; Večeřa et al., 2008), with a HONO/NOx ratio of 0.51 ± 0.18% for fresh plumes (Sun et al., 2020). At CVAO, the observed spikes in $NO_2$ and NO mixing ratio, typically ascribed to shipping plumes, did not correspond with any spikes in

HONO (Fig 1). This suggests that shipping plumes were not a major source of HONO at CVAO, because possibly only aged shipping emissions reached the sampling point during the measurements. At WAO, shipping plumes can be positively identified via sharp spikes in the measured $SO_2$ concentrations, and while such $SO_2$ spikes were observed during the campaign, they did not coincide with any HONO peaks (Fig S11, Supporting information). Furthermore, analysis of the correlations between $SO_2$ and HONO as a function of wind direction revealed that, for sectors from the ocean (NE, N and

NW), the correlation co-efficient was not markedly stronger compared to other sectors.

As noted in the introduction, a range of HONO production mechanisms have been reported in the literature and these processes likely occur, in parallel but to differing degrees of significance, in different boundary layer environments. In the remote marine boundary layer, the observed diurnal trends (and particularly the noontime maximum in [HONO] at CVAO)

point to a dominant source related to sunlight, i.e. photolysis. Recent work has demonstrated that rapid photolysis of particle nitrate can be a significant source of HONO in the marine boundary layer (Kasibhatla et al., 2018; Reed et al., 2017; Ye et al., 2016). Using a 0-D box model, Reed et al. (2017) were only able to reproduce the observed diurnal trends in HONO and NOx at CVAO by including the rapid photolysis of particle nitrate. Although significant uncertainty remains over the importance of nitrate photolysis (Shi et al., 2021), the modelling result of Reed et al suggests that particle nitrate photolysis

may represent the dominant daytime source of HONO at CVAO, as opposed to photochemical conversion of $NO_2$ to HONO on the ocean surface.

## 5 Conclusions

We present measurements of HONO and NOx at two sites located in close proximity to the marine boundary layer: a remote location representative of the open ocean tropical marine boundary layer (Cape Verde, CVAO) and a semi-polluted location

representative of the seas around Europe (Weybourne, WAO). The data are analysed to assess the potential contribution to HONO levels, and hence OH formation, from sea-surface mediated conversion of $NO_2$ to HONO, at night and during the daytime, as posited by Zha et al. (2014). The observed HONO abundance and the HONO/$NO_2$ ratio at the Cape Verde observatory (for which air masses were overwhelming representative of the global remote tropical marine atmosphere) are not consistent with the conversion of $NO_2$ through a dark, sea-surface mechanism. We find an upper limit of $C_{HONO} = 0.0011$

hr$^{-1}$ for the conversion rate coefficient of $NO_2$ to HONO on the sea surface, which is a factor of 6 smaller than the slowest conversion rate obtained from previous studies around the Chinese coast. The situation at Weybourne is more complex, with multiple HONO formation mechanisms likely occurring in parallel; the night-time WAO data cannot be reconciled with an





ocean-surface mechanism producing HONO at the rates observed around China, but the WAO data are consistent with the upper limit on $C_{HONO}$ we obtained from the Cape Verde data. At both locations, the noontime maximum in the diurnal

profile of the HONO/NO$_2$ ratio provides strong evidence for photo-enhanced HONO formation, potentially including both aerosol and sea surface components. The night-time observations reported here can only be reconciled with night-time marine HONO formation rates reported in the literature if there exists substantial regional variability in the rate of a dark, ocean surface NO$_2$-HONO conversion mechanism, which could possibly be explained by variability in the sea-surface microlayer composition. Consequently, caution should be adopted in extrapolating such conversion rates from a given

location to perform regional (and wider) predictions of HONO production, and hence OH levels and oxidation budgets in the MBL. As the marine boundary layer is an important environment globally, better constraint on ocean sources of HONO are critical for understanding global atmospheric oxidative capacity and the lifetime of long-lived species such as CH$_4$ that impact pollution and climate.

**Acknowledgments**

The authors would like to thank Luis Neves, Instituto Nacional de Meteorologia e Geofisica (INMG), Cape Verde, for day-to-day running of the CVAO NOx measurements. The University of Leicester authors thank Roberto Sommariva for his technical and organisational input into the WAO campaign, and Roland Leigh (now EarthSense Systems Ltd) for processing the spectral radiometer data from WAO.

This work was supported by Natural Environment Research Council (NERC) UK through projects ICOZA (Integrated

Chemistry of Ozone in the Atmosphere, NE/K012169/1), SNAABL (Sources of Nitrous Acid in the Atmospheric Boundary Layer, NE/M013545/1 (Birmingham) and NE/M010554/1 (Leicester)), Atmospheric Reactive Nitrogen Cycling over the Ocean (NE/S000518/1), and Assessment of ClNO$_2$ as a missing oxidant in the UK atmosphere (NE/K004069/1).

**Data Availability**

Data used in this study is publicly available from Centre for Environmental Data Analysis (CEDA, www.ceda.ac.uk) and the

EBAS database (Cape Verde data, http://ebas.nilu.no/)

**Author Contributions**

Conceptualization: LRC, WJB. Data curation and Investigation: LRC, LJK, CR, JDL, LDJH, SJB. Formal analysis: LRC, LJK, WJB. Supervision: LJC, JDL, SJB, WJB. Writing – original draft preparation: LRC. Writing- review and editing: All authors.

**Competing interests**

The authors declare that they have no conflict of interest.



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
