# Peer review of "Is the ocean surface a source of HONO in the marine boundary layer?"

_Atmospheric Chemistry and Physics, 2021_

## Author Response (AR2)

Response to editors comments

I found a few small typos to correct:
First line of abstract: layer instead of later.
Line 270: Some kind of punctuation needed after 'observations'?
Line 287: "HONO photolysis frequenices"
Line 362: subscript missing.

*Thank you for spotting these types, these changes have been made to revised submitted version.*

Response to Reviewers

*We thank the reviewers for their considered and thoughtful comments on our manuscript. Please find our response to their comments below.*

**Reviewer 1**

This is an interesting new contribution aiming at exploring the ocean-surface mediated conversion of NO2 to HONO in the coastal marine boundary layer at two contrasting coastal locations, namely at Cape Verde (Atlantic Ocean), representative of the clean remote tropical marine boundary layer, and Weybourne (United Kingdom), representative of semi-polluted Northern European coastal waters. By monitoring, among others, NO2 and HONO, it is shown that the HONO production at these two sites differs significantly from previous reports (a factor of 5 lower). As stated by these authors, these results point to significant geographical variation in the predominant HONO formation mechanisms in marine environments and indicate that caution is required when extrapolating the importance of such mechanisms from individual study locations to assess regional and/or global impacts on oxidizing capacity.

I enjoyed reading this manuscript which is well written and illustrated.

Obviously, the approach used here to detangle whether the NO2 conversion is a source of oceanic HONO is very (or too?) simple, as illustrated by equations 1 and 2. Maybe the reader should be made more aware that this approached oversimplifies the chemistry potentially involved. NO2 reacts quite slowly with water, at a rate dependent on the actual NO2 concentration. Therefore, the first order assumption used to derive equations 1 and 2 may not hold everywhere (even if it unlikely that the second order term may dominate).

*Response:*

As per R5, the overall stoichiometry of the reaction(s) of $NO_2$ with water points to a second order reaction with respect to $NO_2$, even if water is in excess (as two $NO_2$ molecules are consumed). However, there could also be a first order component for HONO production if $NO_2$ is oxidizing hydrocarbons at the surface of the SML (R6). Numerous laboratory studies

have indicated that the kinetics of R5 are primarily dependent on the $NO_2$ concentration, but also on the surface properties (Finlayson-Pitts et al., 2003; Spataro and Ianniello, 2014). The actual mechanism of R5 is not fully understood yet, but is thought to involve the formation of $NO_2$ dimer, $N_2O_4$, that reacts with water to form HONO and $HNO_3$ (Finlayson-Pitts et al., 2003).

We note that Eqn 1 was taken from the literature and was derived empirically using field measurements (Zha et al., 2014). We agree that Eqns 1 and 2 are a simplification of the chemistry likely occurring on the ocean surface, as it neglects the role of surface properties, hence our interest in comparing predicted HONO levels from these equations to our field datasets at CVAO and WAO. To clarify this, we have added the following text in Section 3.3.1 when Eqn 1 is first described:

*"As per R5, the overall reaction between $NO_2$ and water is not first order, however there could be a first order component for HONO production via R6. Hence the empirically derived Eqn 1 is a significant simplification of the chemistry involved as it neglects the role of surface properties but one that reflects the importance of $NO_2$ concentrations on the reaction kinetics (Finlayson-Pitts et al., 2003; Spataro and Ianniello, 2014)."*

Trusting reported values, in this study but also previous ones, it is surprising that the HONO production rates vary to such an extent, and I do regret that section 4 does not discuss more the potential reasons.

The potential role of the SML is briefly discussed when reporting the date from Yu et al (2021), who observed little uptake of NO2 on bulk sea water under dark conditions (γ = 1.6 ×10-8). Indeed, bulk and surface water could exhibit different compositions. Could the chemistry unraveled by the group of Markus Amman (i.e., charge exchange reactions from dissociated phenols to NO2, chemistry occurring at higher pH) be this source of HONO? This could in organic enriched SMLs, which will depend on location and season. And in fact, the fact the seasonality (and therefore the presence or absence of the SML) be one reason of the observed discrepancies between the different studies? Maybe the authors could compare SML maps for the different location and time and sampling to see whether or not such a correlation exists?

*Response:*

We agree that as the SML in enriched with organics this could facilitate the conversion of $NO_2$ to HONO via reaction R6, and possibly via charge exchange reactions from dissociated phenols to $NO_2$ as suggested by the reviewer. However, without information on the composition of the SML at each location during the measurements it is difficult to comment further, as the SML will vary in response to changes in local near-term biological

activity, wind speed and solar radiation (see e.g. (Sabbaghzadeh et al., 2017; Stolle et al., 2020; Wurl et al., 2011).

The geographical/spatial variability in SML composition is not well known (Engel et al., 2017). Modelling work by Wurl et al (2011) suggests that on a global scale the Pacific and Atlantic Oceans between 30° N and 30° S may be more significantly covered with SML than north of 30° N and south of 30° S. Based on this, if HONO production was dependent on the SML, we would expect different $C_{HONO}$ rates to occur at WAO (52.95°N) compared to CVAO (16.864°N) - yet this was not what we observed. The observed similarity in $C_{HONO}$ rates at WAO and CVAO are more in agreement with the findings of Sabbaghzadeh et al. (2017), who observed Atlantic Ocean SML was more uniform and ubiquitous between 50°N and 50°S. Overall, due to the current lack of knowledge on the geographic variability and chemical composition of the SML, it is difficult to explore if discrepancies between the current work and the literature are related to changes in SML.

To clarify the effect of the SML we have added the following text in Section 4.0, line 348:

*"Variability in the observed HONO/NO2 behaviour overnight may reflect geographical differences in the organic composition of the SML, which will vary in response to changes in local near-term biological activity, underlying surface water composition, anthropogenic inputs, wind speed and solar radiation (see e.g. (Sabbaghzadeh et al., 2017; Stolle et al., 2020; Wurl et al., 2011; Wurl and Obbard, 2004).  The geographical/spatial variability in SML composition is not well known (Engel et al., 2017). Modelling work by Wurl et al (2011) suggests that on a global scale the Pacific and Atlantic Oceans between 30° N and 30° S may be more significantly covered with SML than north of 30° N and south of 30° S. Based on this, if HONO production was dependent on the SML, we would expect different $C_{HONO}$ rates to occur at WAO (52.95°N) compared to CVAO (16.864°N) - yet this was not what we observed. The observed similarity in $C_{HONO}$ rates at WAO and CVAO are more in agreement with the findings of Sabbaghzadeh et al. (2017), who observed Atlantic Ocean SML was more uniform and ubiquitous between 50°N and 50°S. Overall, due to the current lack of knowledge on the geographic variability and chemical composition of the SML, it is difficult to explore if discrepancies between the current work and the literature are related to changes in SML. Further work is required to verify whether the SML does or does not facilitate conversion of NO2 to HONO and hence parameterise the variability in the potential night-time NO2-HONO conversion rate on the ocean surface."*

**Reviewer 2**

This manuscript uses two high-quality observational datasets of HONO, NO and NO2 from atmospheric observatories at Cape Verde and Weybourne to test whether the high rates of

conversion of NO2 to HONO in the marine boundary layer reported in other studies are seen at these locations. The results are important and interesting – both during the night and the day, the inferred rates constants for NO2 to HONO conversion at both sites are much smaller than recent reports. The paper is well-written, and should be considered for publication in ACP but could be strengthened by addressing the following comments:

- The authors provide information about the limits of detection of the instruments, but they don't address how observations that are below these limits are included in the analysis. For the Cape Verde site, it seems like there are often low HONO and NOx values and it would be interesting to know how these are treated in the analysis, and what impact their inclusion or exclusion may have on the results.

*Response:*

Data below the detection limit (DL) for HONO or $NO_2$ were not excluded from any analysis, including calculating the mean diurnal trends, to avoid artificially biasing the means high. At CVAO the DL ($2\sigma$) for HONO and $NO_2$ were 0.2 and 8.4 ppt, respectively. For our calculation of rate of conversion of $NO_2$ to HONO ($C_{HONO}$) we used the mean diurnal abundance of $NO_2$ (Section 3.3.1, 35 ppt at night), which is well above the $NO_2$ instrument's DL, indicating the majority of the data was above DL.

To clarify this point, information on the DL of $NO_2$ measurements at CVAO has been added to section 2.2.1:

*"The instrument detection limit ($2\sigma$) for $NO_2$ measurements was 8.4 ppt (Andersen et al., 2020)."*

- The constraints on NO2 to HONO conversion during the day are complicated by the existence of another path of formation (OH + NO). The authors mention briefly that no other paths were included in the calculation, but is there any estimate of how fast the gas phase production mechanism could be?

*Response:*

We can estimate the rate of formation of HONO via OH + NO using reported concentrations of OH from the literature at both sites. Reported noontime levels of OH at CVAO and WAO are $9\times10^6$ (Whalley et al., 2010) and $3\times10^6$ molecules $cm^{-3}$ (Woodward-Massey et al., 2020). Using these OH values, a (second order) OH + NO rate constant of k = $9.7\times10^{-12}$ molecule$^{-1}$ $cm^3$ $s^{-1}$, and mean NO levels at noon (6 and 500 ppt for CVAO and WAO, Fig 3), we can estimate the rate of HONO production via OH + NO at noon. We calculated a rate of 2 and 52 ppt $hr^{-1}$, or $1.3\times10^4$ and $3.6\times10^5$ molecules $cm^{-3}$ $s^{-1}$, for CVAO and WAO respectively. We note that OH levels typically peak at noon and, because this calculation does not consider any sinks (e.g. photolysis), this represents the upper limit to HONO formation from OH + NO.

These values may be compared with the total HONO production rate, which can be estimated from the measured HONO concentration and measured j(HONO) values. This approach assumes steady state for HONO, where photolysis is HONO's only sink, and thus total HONO production $P_{TOT,HONO}= j_{HONO}$ x [HONO]. The corresponding $P_{TOT,HONO}$ values are $1.2 \times 10^5$ and $2.2 \times 10^7$ molecules cm$^{-3}$ s$^{-1}$ for CVAO and WAO respectively. Comparing these totals with the HONO source from the OH + NO reaction, the latter can be seen to account for 11% and 2% (respectively) of the total HONO production at midday, i.e. not negligible, but not such a dominant fraction of the total HONO production that this paper's conclusions re other sources are impacted.

To clarify the role of homogeneous gas-phase production, we have added the following text at line 287:

*"The contribution of R2 to HONO production is estimated at around 10% (midday) using previously measured OH concentrations for CVAO (Whalley et al., 2010)."*

- Because the results at these two sites are so different from other reports, it's interesting to consider what factors might be driving these differences. Is it possible that the reaction is not simply first order in NO2, and thus is higher in more polluted marine boundary layers? If a pseudo-first order rate constant is appropriate, is it likely to be governed by the uptake coefficient of NO2 at the surface (and thus influenced by SML composition) or are there likely to be mass transfer limitations in the atmosphere that restrict the reaction? Do the authors have measurements of a aerodynamic and boundary layer resistance to deposition at either of the sites?

*Response:*

See response to Reviewer 1, comment 1: we agree that (as per R5) the overall stoichiometry of the reaction of $NO_2$ and water is not first order. While the calculated $C_{HONO}$ at WAO was substantially lower than previous work in the East China Sea, the ambient $NO_2$ levels were similar. This would suggest that the rate of HONO production was not related to the level of pollution or $NO_2$, i.e. that higher order effects (with respect to $NO_2$) are not responsible for the discrepancy in results between the locations considered here This is mentioned in the text at lines 319-327:

[revised manuscript text omitted]